# Development and internal validation of prediction models for future hospital care utilization by patients with multimorbidity using electronic health record data

**Marlies Verhoeff**[1,2‡]*, **Janke de Groot**[2☉], **Jako S. Burgers**[3☉], **Barbara C. van Munster**[1☉]

**1** Department of Internal Medicine, University Center of Geriatric Medicine, University Medical Center Groningen, Groningen, The Netherlands, **2** Knowledge Institute of the Federation of Medical Specialists, Utrecht, the Netherlands, **3** Faculty of Health, Medicine and Life Sciences (FHML), Maastricht University, Maastricht, the Netherlands

☉ These authors contributed equally to this work.
‡ These authors share first authorship on this work.
* m.verhoeff@umcg.nl

**Data Availability Statement:** Data cannot be shared publicly because of privacy reasons: the data contains patient sensitive information and is

## Abstract

### Objective

To develop and internally validate prediction models for future hospital care utilization in patients with multiple chronic conditions.

### Design

Retrospective cohort study.

### Setting

A teaching hospital in the Netherlands (542 beds)

### Participants

All adult patients (n = 18.180) who received care at the outpatient clinic in 2017 for two chronic diagnoses or more (including oncological diagnoses) and who returned for hospital care or outpatient clinical care in 2018. Development and validation using a stratified random split-sample (n = 12.120 for development, n = 6.060 for internal validation).

### Outcomes

≥2 emergency department visits in 2018, ≥1 hospitalization in 2018 and ≥12 outpatient visits in 2018.

### Statistical analysis

Multivariable logistic regression with forward selection.

bound to national privacy regulation. Data are available from the Gelre Institutional Data Access / Ethics Committee (contact via wetenschap@gelre.nl) for researchers who meet the criteria for access to confidential data.

**Funding:** The author(s) received no specific funding for this work.

**Competing interests:** The authors have declared that no competing interests exist.

## Results

Evaluation of the models' performance showed c-statistics of 0.70 (95% CI 0.69–0.72) for the hospitalization model, 0.72 (95% CI 0.70–0.74) for the ED visits model and 0.76 (95% 0.74–0.77) for the outpatient visits model. With regard to calibration, there was agreement between lower predicted and observed probability for all models, but the models overestimated the probability for patients with higher predicted probabilities.

## Conclusions

These models showed promising results for further development of prediction models for future healthcare utilization using data from local electronic health records. This could be the first step in developing automated alert systems in electronic health records for identifying patients with multimorbidity with higher risk for high healthcare utilization, who might benefit from a more integrated care approach.

## Introduction

The prevalence of multimorbidity (defined as having two or more chronic conditions) is increasing [1]. Kingston et al. (2018) predicted that by 2035 67.8% of the adults in the UK aged over 65 years will be living with multimorbidity [2]. An increasing prevalence of multimorbidity puts pressure on current healthcare systems, as hospital organizations are mostly providing disease-specific care that is generally delivered by separate disciplines or medical specialties [3,4]. Compared to patients with single chronic conditions, patients with multimorbidity have a higher risk of experiencing fragmented care, possibly resulting in suboptimal outcomes [4–8]. Fragmentation of care, especially with a lack of care coordination, can lead to adverse outcomes such as over- or undertreatment, unnecessary diagnostics and medication-interactions [9–12]. If undetected, these consequences can result in unnecessary and potentially preventable healthcare utilization, such as emergency department (ED) visits, hospitalizations and outpatient visits [13].

Several (inter)national healthcare organizations suggest that quality of care for patients with multimorbidity might improve with a more integrated care approach, for example by organizing better coordination and more tailoring of care [14]. This approach might also reduce the risk and related costs of adverse outcomes and decrease preventable future healthcare utilization, like emergency department visits, acute hospitalization and unnecessary outpatient visits [14–16]. Nevertheless, to allocate healthcare resources in a way that is both feasible and sustainable, healthcare professionals should identify patients with multimorbidity that might benefit most from a more integrated care approach, such as those with a high modifiable risk for adverse outcomes or a high risk of frequent or acute healthcare utilization [14,17]. Several studies found that healthcare utilization as well as high costs are associated with numerous disease-related, patient-related and healthcare-related factors [18–23]. Because of this multifactorial association, it is difficult for individual healthcare professionals to quickly recognize patients with multimorbidity at high risk for future frequent or acute healthcare utilization that potentially could (partially) be prevented with a more integrated care approach.

A risk screening tool might aid healthcare professionals in identifying patients who might benefit most from an integrated care approach. In other fields, several risk screening tools are available, e.g. in cardiovascular risk management and the diagnostic pathway of deep-vein

thrombosis, that combine several patient-related or disease-related factors to support health-care professionals' decisions when dealing with individual patients [24,25]. Normally, the healthcare professional collects data on risk factors to calculate the risk for the individual patient and tailors the treatment strategy based on this risk. The registration of data in the Electronic Health Record (EHR) offers opportunities to develop, integrate and automate the data collection and calculation of an individual patient's risk for specific outcomes, such as future healthcare utilization, using the registered individual patient data [26–28]. Therefore, the aim of this study was to develop, validate and evaluate the performance of prediction models for (1) ≥2 emergency department visits, (2) ≥1 acute hospitalization and (3) ≥12 outpatient visits in patients with multimorbidity, based on administrative EHR data.

## Methods

Our study is a retrospective cohort study of a large hospital population of patients with multi-morbidity. We used data on the population's demographics and healthcare utilization in 2017 to develop and internally validate three prediction models for healthcare utilization outcomes in 2018. We followed the recommendations of the Transparent Reporting of a multivariable prediction model for Individual Prognosis Or Diagnosis (TRIPOD) statement for this article [29].

### Source of data and population

The data used for this study was administrative EHR data on all adult patients with multiple chronic conditions who visited the outpatient clinic of Gelre hospital in Apeldoorn, a middle-large teaching hospital in the Netherlands, in 2017 and 2018. We included all patients who:

■ were aged 18 years or older;

■ had received outpatient clinical care for multimorbidity, defined as at least two chronic con-ditions, in 2017. Both chronic and oncologic diagnoses were considered chronic conditions;

■ had received hospital care for at least one diagnosis in 2018.

The local institutional review board approved the anonymous use of these data for research purposes and a waiver of consent (Local ethics committee Gelre ziekenhuizen (Gelre LTC) number 2019_02).

In the Netherlands, hospital care is coded and billed using billing codes that include diagno-sis and treatment combinations (DTCs). These DTCs contain information about the diagnosis, including an International Classification of Diseases and Related Health Problems 10 (ICD-10) code. The DTC data also contains information about the location, time, type and number of care activities linked to the specific diagnoses [30]. The diagnoses were classified into 259 clinically relevant diagnosis groups with the use of Clinical Classifications Software (CCS) for ICD-10-PCS, which was developed by the Agency for Healthcare Research and Quality (AHRQ) [31]. The diagnoses from the CCS classification were categorized by Dutch Hospital Data in acute, chronic, elective, oncological and other diagnoses.

### Outcomes

We included three types of healthcare utilization:

• Acute hospitalization(s);

• Multiple emergency department (ED) visits;

- A high number of outpatient visits.

Acute hospitalization(s) was defined as one or more acute hospitalizations in 2018. Acute hospitalization is a major potential adverse event that, when caused by the consequences of care fragmentation, might be preventable.

Multiple ED visits were defined as two or more ED visits in 2018, which is consistent with other definitions of frequent ED visits [32]. One ED visit can happen to anyone, but more than one ED visit could suggest that there is a more chronic cause for acute care utilization. If care fragmentation and the described consequences are present in these patients, some of the ED visits might be preventable.

A high number of outpatient visits was defined as twelve or more outpatient visits in 2018, which is on average one outpatient visit per month. A recent study that developed prediction models for high care need in patients with multimorbidity from the primary care perspective also used twelve or more contacts with the general practitioner as the cut-off point [33].

## Predictors

We selected candidate predictor variables based on existing literature and clinical expertise. The demographic characteristics consisted of age, sex, and socio-economic status [18–23,34]. Socio-economic status was based on ZIP code and classified as 'low', 'medium' and 'high' based on information from the Central Bureau of Statistics Netherlands [35]. The healthcare utilization characteristics included number of chronic diagnoses, number of acute diagnoses and number of medical specialties involved in the patients' hospital care in 2017. Additionally, we calculated and included the number of outpatient visits, number of acute hospitalizations, number of inpatient days, number of ICU days and number of emergency department visits, using the information on care activities in 2017. We imputed missing values with the mean of non-missing cases [36].

## Statistical analysis

We used R 3.6.1, Rstudio 1.2.5001 and the *pROC* (*v1.17.0.*1) package for the statistical analysis and the *ggplot2* (*v3.3.3*) package for visualization [37–40]. For continuous and discrete numerical variables, we checked the linearity of the log odds by categorizing the variable into n groups and analyzing the association with the outcome with the use of (n-1) dummy variables reported by tables and plots. If the association was approximately linear, the variable was included as continuous variable in the model. If the association was defined as non-linear, the variable was split into groups and included in the model as a categorical variable. The number of categories for a variable was determined with the following steps:

1. The dataset was divided into quantiles (starting with quartiles, followed by either terciles or quintiles/sextiles depending on the percentile distribution of the variables)

2. We assessed the quantile's cut-off points. If the cut-off points were deemed clinically relevant, these cut-off points were used. The cut-off points were clinically relevant if they had been used in prior research or if they were meaningful based on clinical practicality and meaningfulness (determined by the researchers). Quantile's cut-off points were rounded to clinically relevant cut-off points if possible, to stay as close as possible to the quantile distribution of patients.

3. If the groups and cut-off points produced by the quantiles were not deemed clinically relevant, the cut-off points and number of groups were determined based on clinical practicality and meaningfulness (determined by the researchers). A minimum of 50 events per

group was used. Moreover, the overlap of the odds ratios and confidence intervals for the groups were assessed. A solution with a number of groups with no overlap in confidence intervals was preferred, if possible.

For the log odds of the number of acute diagnoses and number of emergency department visits, a linear relationship with the log odds of the outcome was found to be a good approximation after assessment using the prior described steps. For age, number of chronic/oncologic diagnoses, number of specialties, number of outpatient visits, number of acute hospitalizations, number of inpatient days, number of ICU days, and number of therapeutic care activities, groups were prepared using the methods as earlier described, with the following cut-off-points:

- Age: 55, 65, and 75 years;

- Number of chronic/oncologic diagnoses in 2017: 2, 3, 4, 5 and 6 chronic/oncologic diagnoses;

- Number of specialties involved in 2017: 2, 3, 4, 5 and 6 specialties;

- Number of outpatient visits in 2017: 5 and 8 outpatient visits.

- Number of acute hospitalizations in 2017: 1 and 2 acute hospitalization(s)

- Number of inpatient days in 2017: 1, 4 and 8 inpatient days.

## Building the prediction models

We used multivariable logistic regression with forward selection. For each step we started by adding the candidate predictor with the lowest p-value to the model. We stopped adding new variables to the model when all remaining candidate predictors had a p-value < 0.05. We assessed internal validity with a weighted split-sample procedure. We randomly split the sample into development sets (two third of the sample size) and validation sets (one third of the sample size) aiming for a minimum of 50 cases per predictor group in the development sets [41]. We assured the same distribution for every outcome in both sets by first grouping the data by outcome. To evaluate performance of each model we examined discrimination with a ROC-curve and calculated the c-statistic (AUC) using the *pROC* package and examined calibration by plotting the calibration curve.

## Results

### Population characteristics

Overall, 18180 patients were included (S1 Fig). Table 1 shows the general, disease and care characteristics in 2017. Median age of the population was 68.0 years (IQR 48.1–87.8 years). 61.6% of the included patients had two diagnoses, 24.4% had three diagnoses and the remaining patients had four or more chronic and/or oncologic diagnoses for which they had used hospital care. With regard to the outcomes in 2018, 2257 patients (12.4%) had at least one hospitalization in 2018, 1258 (6.9%) had two or more ED visits in 2018 and 1293 patients (7.1%) had at least 12 outpatient visits in 2018. After checking for linearity between the log odds of the outcomes and the candidate predictors, only the number of acute diagnoses in 2017 and number of ED visits in 2017 were included without groups. All other candidate predictors showed a non-linear relationship to the outcomes and were split into groups. The characteristics of the validation datasets were similar to those of the derivations datasets (S1 Table). The estimated socio-economic status had missing values (n = 47) due to missing socio-economic status information for a number of ZIP codes. These missing values were imputed with the mean of non-missing cases.

**Table 1. Population characteristics in 2017.**

| Variable | Total population (n = 18180) |
|---|---|
| **General characteristics** | |
| Age, median | 68.0 (48.1–87.8) |
| Age (groups) | 4132 (22.7) |
| *18–54 years* | 3585 (19.7) |
| *55–64 years* | 5369 (29.5) |
| *65–74 years* | 5094 (28.0) |
| *75–98 years* | |
| Sex, female, n (%) | 10289 (56.6) |
| Socioeconomic status, n (%) | |
| *Low* | 7243 (39.8) |
| *Middle* | 7244 (39.9) |
| *High* | 3693 (20.3) |
| **Disease characteristics** | |
| Chronic/oncologic diagnoses, median (IQR) | 2 (2–3) |
| Chronic/oncologic diagnoses (groups) n(%) | 11203 (61.6) |
| *2 chronic/oncologic diagnoses* | 4433 (24.4) |
| *3 chronic/oncologic diagnoses* | 1653 (9.1) |
| *4 chronic/oncologic diagnoses* | 575 (3.2) |
| *5 chronic/oncologic diagnoses* | 316 (1.7) |
| *≥6 chronic/oncologic diagnoses* | |
| Acute diagnoses, median (IQR) | 0 (0–1) |
| **Hospital care characteristics** | |
| Medical specialties involved, median (IQR) | 3 (2–5) |
| Medical specialties involved (groups), n(%) | 5559 (30.6) |
| *2 specialties* | 5817 (32.0) |
| *3 specialties* | 3662 (20.1) |
| *4 specialties* | 1797 (9.9) |
| *5 specialties* | 1345 (7.4) |
| *≥6 specialties* | |
| Outpatient visits, median (IQR) | 6 (2–10) |
| Outpatient visits (groups), n (%) | 6337 (34.9) |
| *2–4 visits* | 6176 (34.0) |
| *5–7 visits* | 5667 (31.2) |
| *≥8 visits* | |
| Acute hospitalizations, median (IQR) | 0 (0–0) |
| Acute hospitalizations (groups) | 14847 (81.7) |
| *0 acute hospitalizations* | 2479 (13.6) |
| *1 acute hospitalization* | 854 (4.7) |
| *≥2 acute hospitalizations* | |
| Inpatient days, median (IQR) | 0 (0–2) |
| Inpatient days (groups) | 13402 (73.7) |
| *0 inpatient days* | 1504 (8.3) |
| *1–3 inpatient days* | 1531 (8.4) |
| *4–7 inpatient days* | 1743 (9.5) |
| *≥8 inpatient days* | |
| ICU days, median (IQR) | 0 (0–0) |
| Patients with at least 1 ICU admission, n (%) | 265 (1.4) |
| Emergency department visits, median (IQR), visits | 0 (0–1) |

## Model 1: Predicting at least one hospitalization in 2018 (Table 2)

A higher age in 2017 was associated with at least one hospitalization in 2018. With age group 18–54 years as reference, the OR increased per age group, from 1.53 (95% CI 1.25–1.87) for the age 55–64 years to 2.54 (95% CI 2.13–3.04) for the age 75 or more years. In the univariable analysis, the number of chronic/oncologic diagnoses showed significant associations with at

**Table 2. Results (development data) for outcome '≥1 hospitalization(s) in 2018'.**

| Variable | At least 1 hospitalization in 2018 (n = 1505) | No hospitalization in 2018 (n = 10615) | Univariable Odds Ratio (95% CI) | Multivariable Odds Ratio (95% CI) | p-value |
|---|---|---|---|---|---|
| **General characteristics (2017)** | | | | | |
| Age group, n (%) | | | | | |
| *18–54 years* | 192 (12.8) | 2603 (24.5) | 1 (ref) | 1 (ref) | <0.0001 |
| *55–64 years* | 264 (17.5) | 2143 (20.2) | 1.67 (1.38–2.03) | 1.53 (1.25–1.87) | <0.0001 |
| *65–74 years* | 434 (28.8) | 3135 (29.5) | 1.88 (1.57–2.25) | 1.67 (1.39–2.02) | <0.0001 |
| *75–98 years* | 615 (40.9) | 2734 (25.8) | 3.05 (2.58–3.63) | 2.54 (2.13–3.04) | |
| Sex, female, n (%) | 746 (49.6) | 6143 (57.9) | 0.72 (0.64–0.80) | 0.80 (0.71–0.89) | 0.0001 |
| Socioeconomic status, n (%) | | | | | |
| *Low* | 630 (41.9) | 4185 (39.4) | 1 (ref) | 1 (ref) | 0.0610 |
| *Middle* | 595 (39.5) | 4244 (40.0) | 0.93 (0.83–1.05) | 0.89 (0.78–1.01) | 0.0245 |
| *High* | 280 (18.6) | 2186 (20.6) | 0.85 (0.73–0.99) | 0.84 (0.71–0.98) | |
| **Disease characteristics (2017)** | | | | | |
| Chronic/oncologic diagnoses, n (%) | | | | | |
| *2 chronic/oncologic diagnoses* | 739 (49.1) | 6748 (63.6) | 1 (ref) | 1 (ref) | 0.1228 |
| *3 chronic/oncologic diagnoses* | 404 (26.8) | 2560 (24.1) | 1.44 (1.27–1.64) | 1.12 (0.97–1.28) | 0.0028 |
| *4 chronic/oncologic diagnoses* | 201 (13.4) | 862 (8.1) | 2.13 (1.79–2.52) | 1.34 (1.10–1.62) | 0.4017 |
| *5 chronic/oncologic diagnoses* | 82 (5.5) | 309 (2.9) | 2.42 (1.87–3.11) | 1.13 (0.85–1.49) | 0.0005 |
| *≥6 chronic/oncologic diagnoses* | 79 (5.3) | 136 (1.3) | 5.30 (3.97–7.05) | 1.80 (1.29–2.50) | |
| Acute diagnoses, median (IQR), diagnoses | 0 (0–1) | 0 (0–1) | 1.67 (1.57–1.78) | 1.09 (1.01–1.18) | 0.0296 |
| **Hospital care characteristics (2017)** | | | | | |
| Medical specialties involved, n (%) | | | | | |
| *2 specialties* | 312 (20.7) | 3440 (32.4) | 1 (ref) | | |
| *3 specialties* | 424 (28.2) | 3458 (32.6) | 1.35 (1.16–1.58) | | |
| *4 specialties* | 327 (21.7) | 2069 (19.5) | 1.74 (1.48–2.05) | | |
| *5 specialties* | 231 (15.4) | 978 (9.2) | 2.60 (2.16–3.13) | | |
| *≥6 specialties* | 211 (14.0) | 670 (6.3) | 3.47 (2.86–4.21) | | |
| Outpatient visits, n(%) | | | | | |
| *2–4 visits* | 366 (24.32) | 3884 (36.6) | 1 (ref) | 1 (ref) | 0.8441 |
| *5–7 visits* | 442 (29.37) | 3685 (34.7) | 1.27 (1.10–1.47) | 1.02 (0.87–1.18) | 0.0065 |
| *≥8 visits* | 697 (46.3) | 3046 (28.7) | 2.43 (2.12–2.78) | 1.26 (1.07–1.50) | |
| Acute hospitalizations, n (%) | | | | | |
| *No acute hospitalizations* | 943 (62.7) | 8908 (83.9) | 1 (ref) | 1 (ref) | 0.0679 |
| *1 acute hospitalization* | 345 (22.9) | 1359 (12.8) | 2.40 (2.09–2.74) | 1.23 (0.99–1.55) | 0.0100 |
| *≥2 acute hospitalizations* | 217 (14.4) | 348 (3.3) | 5.89 (4.90–7.06) | 1.55 (1.11–2.16) | |

*(Continued)*

**Table 2.** (Continued)

| Variable | At least 1 hospitalization in 2018 (n = 1505) | No hospitalization in 2018 (n = 10615) | Univariable Odds Ratio (95% CI) | Multivariable Odds Ratio (95% CI) | p-value |
|---|---|---|---|---|---|
| Inpatient days, n(%) | | | | | |
| *No inpatient days* | 821 (54.6) | 8085 (76.2) | 1 (ref) | 1 (ref) | 0.4469 |
| *1–3 inpatient days* | 142 (9.4) | 870 (8.2) | 1.61 (1.32–1.94) | 1.09 (0.87–1.37) | 0.1423 |
| *4–7 inpatient days* | 188 (12.5) | 844 (8.0) | 2.19 (1.84–2.60) | 1.20 (0.94–1.51) | 0.0033 |
| *≥8 inpatient days* | 354 (23.5) | 816 (7.7) | 4.27 (3.70–4.93) | 1.47 (1.13–1.89) | |
| Patients with at least 1 ICU admission, n (%) | 27 (1.8) | 139 (1.3) | 1.38 (0.89–2.05) | | |
| Emergency department visits, median (IQR), visits | 1 (0–2) | 0 (0–1) | 1.62 (1.54–1.70) | 1.23 (1.15–1.33) | <0.0001 |

least one hospitalization in 2018, but in the multivariable analysis these associations disappeared, except for the group with six or more chronic/oncologic diagnoses (OR 1.80, 95% CI 1.29–2.50). Moreover, variables of acute care utilization in 2017 where significant predictors in the multivariable model for hospitalization in 2018. Patients with two or more hospitalizations in 2017 had 1.55 higher odds (95% CI 1.11–2.16) to have at least one hospitalization in 2018 compared to patients without a hospitalization in 2017. Moreover, every ED visit in 2017 led to 1.23 higher odds (95% CI 1.15–1.33) of having at least one hospitalization in 2018. Compared to patients with no inpatient days, patients with eight or more inpatient days had 1.47 higher odds (95%CI 1.13–1.89) of having at least one hospitalization in 2018.

## Model 2: Predicting 2 or more ED visits in 2018 (Table 3)

Age in 2017, number of outpatient visits in 2017 and variables of acute healthcare utilization in 2017 were predictors for 2 or more ED visits in 2018. The association of number of chronic/oncologic diagnoses in 2017 with 2 or more ED visits in 2018 in the univariable analysis disappeared in the multivariable analysis, with exception of the group with six or more chronic diagnoses. In the multivariable model, patients with 4–7 or ≥8 inpatient days had 1.22 (95% CI 0.99–1.51) and 1.72 (95% CI 1.37–2.17) higher odds of visiting the ED 2 or more times in 2018, respectively, compared to patients with no inpatient days. Every ED visit in 2017 led to 1.49 (95% CI 1.39–1.61) higher odds of visiting the ED 2 or more times in 2018. Moreover, patients with eight or more outpatient visits in 2017 had an OR 1.80 (1.29–250) of visiting the ED twice or more in 2018.

## Model 3: Predicting 12 or more outpatient visits in 2018 (Table 4)

Age, number of chronic/oncologic diagnoses, higher numbers of involved medical specialties and number of outpatient and ED visits were significant predictors of the outcome in the multivariable model for 12 or more outpatient visits in 2018. The number of outpatient visits in 2017 was the strongest predictor of 12 or more outpatient visits in 2018. The OR was 2.13 (95% CI 1.52–2.61) for five to seven outpatient visits and 4.76 (95% CI 3.63–6.29) for eight or more outpatient visits compared to patients with two to four outpatient visits.

## Performance and internal validation of the models

Evaluation of the models' performance showed c-statistics of 0.70 (95% CI 0.69–0.72) for the hospitalization model, 0.72 (95% CI 0.70–0.74) for the ED visits model and 0.75 (95% CI 0.73–

**Table 3. Results (development data) for outcome '≥2 ED visits in 2018'.**

| Variable | ≥2 ED visits (n = 839) | No or 1 ED visit (n = 11.282) | Univariable Odds Ratio (95% CI) | Multivariable Odds Ratio (95% CI) | p-value |
|---|---|---|---|---|---|
| **General characteristics (2017)** | | | | | |
| Age group, n (%) | | | | | |
| 18–54 years | 139 (16.6) | 2547 (22.6) | 1 (ref) | 1 (ref) | 0.4293 |
| 55–64 years | 142 (16.9) | 2186 (19.4) | 1.19 (0.94–1.51) | 1.11 (0.86–1.42) | 0.0252 |
| 65–74 years | 264 (31.5) | 3443 (30.5) | 1.41 (1.14–1.74) | 1.29 (1.03–1.61) | 0.0019 |
| 75–98 years | 294 (35.0) | 3106 (27.5) | 1.73 (1.41–2.14) | 1.42 (1.14–1.77) | |
| Sex, female, n (%) | 443 (52.8) | 6373 (56.5) | 0.86 (0.75–0.99) | | |
| Socioeconomic status, n (%) | | | | | |
| Low | 351 (41.8) | 4449 (39.4) | 1 (ref) | | |
| Middle | 337 (40.2) | 4554 (40.4) | 0.94 (0.80–1.10) | | |
| High | 151 (18.0) | 2279 (20.2) | 0.84 (0.69–1.02) | | |
| **Disease characteristics (2017)** | | | | | |
| Chronic/oncologic diagnoses, n (%) | | | | | |
| 2 chronic/oncologic diagnoses | 389 (46.4) | 7005 (62.1) | 1 (ref) | 1 (ref) | 0.5799 |
| 3 chronic/oncologic diagnoses | 222 (26.5) | 2780 (24.6) | 1.44 (1.21–1.70) | 1.05 (0.88–1.26) | 0.2324 |
| 4 chronic/oncologic diagnoses | 116 (13.8) | 988 (8.8) | 2.11 (1.69–2.62) | 1.16 (0.91–1.47) | 0.6349 |
| 5 chronic/oncologic diagnoses | 53 (6.3) | 346 (3.1) | 2.76 (2.01–3.72) | 1.09 (0.76–1.52) | 0.0006 |
| ≥6 chronic/oncologic diagnoses | 59 (7.0) | 163 (1.4) | 6.52 (4.72–8.88) | 1.90 (1.31–2.72) | |
| Acute diagnoses, median (IQR) | 1 (0–2) | 0 (0–1) | 1.79 (1.66–1.92) | 1.11 (1.01–1.22) | 0.0271 |
| **Hospital care characteristics (2017)** | | | | | |
| Medical specialties involved, n (%) | | | | | |
| 2 specialties | 164 (19.5) | 3504 (31.1) | 1 (ref) | | |
| 3 specialties | 215 (25.6) | 3646 (32.3) | 1.26 (1.02–1.55) | | |
| 4 specialties | 166 (19.8) | 2285 (20.2) | 1.55 (1.24–1.94) | | |
| 5 specialties | 134 (16.0) | 1083 (9.6) | 2.64 (2.08–3.35) | | |
| ≥6 specialties | 160 (19.1) | 767 (6.8) | 4.46 (3.54–5.62) | | |
| Outpatient visits, n(%) | | | | | |
| 2–4 visits | 150 (17.9) | 4042 (35.8) | 1 (ref) | 1 (ref) | 0.0056 |
| 5–7 visits | 239 (28.5) | 3835 (34.0) | 1.68 (1.36–2.07) | 1.36 (1.09–1.69) | <0.0001 |
| ≥8 visits | 450 (53.6) | 3405 (30.2) | 3.56 (2.95–4.32) | 1.72 (1.37–2.17) | |
| Acute hospitalizations, n (%) | | | | | |
| No acute hospitalizations | 495 (59.0) | 9350 (82.9) | 1 (ref) | | |
| 1 acute hospitalization | 192 (22.9) | 1505 (13.3) | 2.41 (2.02–2.87) | | |
| ≥2 acute hospitalizations | 152 (18.1) | 427 (3.8) | 6.72 (5.46–8.25) | | |
| Inpatient days, n(%) | | | | | |
| No inpatient days | 427 (50.9) | 8427 (74.7) | 1 (ref) | 1 (ref) | 0.9862 |
| 1–3 inpatient days | 75 (8.9) | 942 (8.3) | 1.57 (1.21–201) | 1.00 (0.76–1.29) | 0.0748 |
| 4–7 inpatient days | 112 (13.3) | 926 (8.2) | 2.39 (1.91–2.96) | 1.24 (0.97–1.57) | 0.0086 |
| ≥8 inpatient days | 225 (26.8) | 987 (8.8) | 4.50 (3.77–5.35) | 1.37 (1.08–1.73) | |
| Patients with at least 1 ICU admission, n (%) | 29 (3.5) | 152 (1.3) | 2.62 (1.72–3.86) | | |
| Emergency department days, median (IQR) | 1 (1–3) | 0 (0–1) | 1.81 (1.72–1.91) | 1.49 (1.39–1.61) | <0.0001 |

0.76) for the outpatient visits model. The c-statistics in two validation sets were almost similar to the c-statistics in the development sets: 0.69 (95% CI 0.67–0.71) for the hospitalization model and 0.75 (95% CI 0.73–0.78) for the outpatient visits model. The model predicting two or more ED visits performed less in the validation set with a c-statistic of 0.67 (95% CI 0.64–0.70). The full prognostic models including intercept and model performance measures for the development and validation sets are included in supplementary tables (see S2–S4 Tables). The models' calibration curves (Fig 1) show that there was agreement between lower predicted and observed probability for hospitalization and ED visits, but that the models overestimated the probability for patients with higher predicted probabilities. For the outpatient visit model

**Table 4. Results (development data) for outcome '≥12 outpatient visits in 2018'.**

| Variable | ≥12 outpatient visits in 2018 (n = 862) | Less than 12 outpatient visits in 2018 (n = 11258) | Univariate Odds Ratio (95% CI) | Multivariable Odds Ratio (95% CI) | P Value |
|---|---|---|---|---|---|
| **General characteristics (2017)** | | | | | |
| Age group, n (%) | | | | | |
| 18–54 years | 127 (14.7) | 2691 (23.9) | 1 (ref) | 1(ref) | 0.0034 |
| 55–64 years | 173 (20.1) | 2194 (19.5) | 1.67 (1.32–2.12) | 1.44 (1.13–1.83) | <0.0001 |
| 65–74 years | 299 (34.7) | 3258 (28.9) | 1.94 (1.57–2.42) | 1.58 (1.27–1.98) | 0.0109 |
| 75–98 years | 263 (30.5) | 3115 (27.7) | 1.79 (1.44–2.23) | 1.34 (1.07–1.69) | |
| Sex, female, n (%) | 451 (52.3) | 6426 (57.1) | 0.83 (0.72–0.95) | | |
| Socioeconomic status, n (%) | | | | | |
| Low | 361 (41.9) | 4530 (40.2) | 1 (ref) | | |
| Middle | 338 (39.2) | 4489 (40.0) | 0.94 (0.81–1.10) | | |
| High | 163 (18.9) | 2239 (19.9) | 0.91 (0.75–1.10) | | |
| **Disease characteristics (2017)** | | | | | |
| Chronic/oncologic diagnoses, n (%) | | | | | |
| 2 chronic/oncologic diagnoses | 321 (37.2) | 7185 (63.8) | 1 (ref) | 1 (ref) | 0.0018 |
| 3 chronic/oncologic diagnoses | 262 (30.4) | 2663 (23.7) | 2.20 (1.86–2.61) | 1.34 (1.12–1.62) | 0.0035 |
| 4 chronic/oncologic diagnoses | 147 (17.1) | 957 (8.5) | 3.44 (2.79–4.22) | 1.43 (1.12–1.81) | 0.0063 |
| 5 chronic/oncologic diagnoses | 73 (8.5) | 310 (2.8) | 5.27 (3.97–6.93) | 1.57 (1.13–2.15) | 0.0001 |
| ≥6 chronic/oncologic diagnoses | 59 (6.8) | 143 (1.3) | 9.24 (6.64–12.70) | 2.17 (1.48–3.17) | |
| Acute diagnoses in 2017, median (IQR), diagnoses | 0 (0–1) | 0 (0–1) | 1.52 (1.41–1.63) | | |
| **Hospital care characteristics (2017)** | | | | | |
| Medical specialties involved, n (%) | | | | | |
| 2 specialties | 123 (14.3) | 3644 (32.4) | 1 (ref) | 1 (ref) | 0.5489 |
| 3 specialties | 188 (21.8) | 3641 (32.3) | 1.53 (1.21–1.93) | 0.93 (0.72–1.19) | 0.8433 |
| 4 specialties | 198 (23.0) | 2238 (19.9) | 2.62 (2.08–3.31) | 1.03 (0.79–1.35) | 0.0546 |
| 5 specialties | 168 (19.5) | 1041 (9.2) | 4.78 (3.76–6.10) | 1.34 (1.00–1.80) | 0.0047 |
| ≥6 specialties | 185 (21.5) | 694 (6.2) | 7.90 (6.21–10.08) | 1.59 (1.15–2.20) | |
| Outpatient visits, n(%) | | | | | |
| 2–4 visits | 90 (10.4) | 4180 (37.1) | 1 (ref) | 1 (ref) | <0.0001 |
| 5–7 visits | 212 (24.6) | 3904 (34.7) | 2.52 (1.97–3.25) | 2.13 (1.52–2.61) | <0.0001 |
| ≥8 visits | 560 (65.0) | 3174 (28.2) | 8.19 (6.56–10.35) | 4.76 (3.63–6.29) | |
| Acute hospitalizations, n (%) | | | | | |
| No acute hospitalizations | 586 (68.0) | 9328 (82.9) | 1 (ref) | | |
| 1 acute hospitalization | 168 (19.5) | 1456 (12.9) | 1.84 (1.53–2.19) | | |
| ≥2 acute hospitalizations | 108 (12.5) | 474 (4.2) | 3.63 (2.88–4.53) | | |

*(Continued)*

**Table 4.** (Continued)

| Variable | ≥12 outpatient visits in 2018 (n = 862) | Less than 12 outpatient visits in 2018 (n = 11258) | Univariate Odds Ratio (95% CI) | Multivariable Odds Ratio (95% CI) | P Value |
|---|---|---|---|---|---|
| Inpatient days, n(%) | | | | | |
| *No inpatient days* | 486 (56.4) | 8490 (75.4) | 1 (ref) | | |
| *1–3 inpatient days* | 90 (10.4) | 900 (8.0) | 1.75 (1.37–2.20) | | |
| *4–7 inpatient days* | 115 (13.3) | 906 (8.0) | 2.22 (1.78–2.74) | | |
| *≥8 inpatient days* | 171 (19.8) | 962 (8.5) | 3.11 (2.57–3.73) | | |
| Patients with at least 1 ICU admission, n (%) | 28 (3.2) | 149 (1.3) | 2.50 (1.63–3.71) | | |
| Emergency department days, median (IQR), days | 0 (0–1) | 0 (0–1) | 1.45 (1.37–1.53) | 1.13 (1.06–1.20) | <0.0001 |

there was good agreement, with a slight underestimation of the probability in the patients with intermediate predicted probability and an overestimation of the probability in the patients with a higher predicted probability.

### Integrating prediction models into the EHR

An individual patient's risk can be calculated using the regression coefficients in the supplementary tables (see S2–S4 Tables). Fig 2 shows an example of how the calculated predicted risk, including the probability percentile (top-X% risk group, see S2 Fig) for the three outcomes for new fictive patients, could be reported to an individual healthcare professional.

### Discussion

The aim of this study was to develop and validate prediction models for future (1) ≥2 emergency department visits, (2) ≥1 acute hospitalization and (3) ≥12 outpatient visits in patients with multimorbidity, using existing administrative EHR data. Our results suggest that local administrative data from the EHR can be used to locally develop and validate reasonable performing prediction models for these outcomes. All prediction models also performed reasonably well in the validation sets (see S2–S4 Tables). The predicted and actual probabilities show good agreement in each model, but show a tendency to overestimate the actual probability in the higher risk groups for ≥1 hospitalization and ≥2 ED visits.

In line with other research, our study shows that administrative data from the EHR can be used to develop reasonable prediction models for healthcare utilization [42]. In a systematic review by Wallace et al. (2014) the best performing models to predict acute hospitalization using administrative or clinical record data had similar c-statistics in development studies ranging from 0.68 to 0.83 [42]. Hudon et al. (2020) developed prediction models to predict four or more ED visits and reported c-statistics of 0.76 and 0.79 [43]. Compared to these models, our acute unplanned care models scored well. However, we were unable to compare calibration to these models, because these studies did not show calibration curves for their data.

To our knowledge, there are no studies that developed prediction models to predict high numbers of outpatient visits, but models using administrative data from primary care predicting persistent frequent attendance and ≥12 general practitioner visits reported c-statistics of respectively 0.67 and 0.83, which is consistent with the c-statistic of 0.75 in our validation set [33,44].

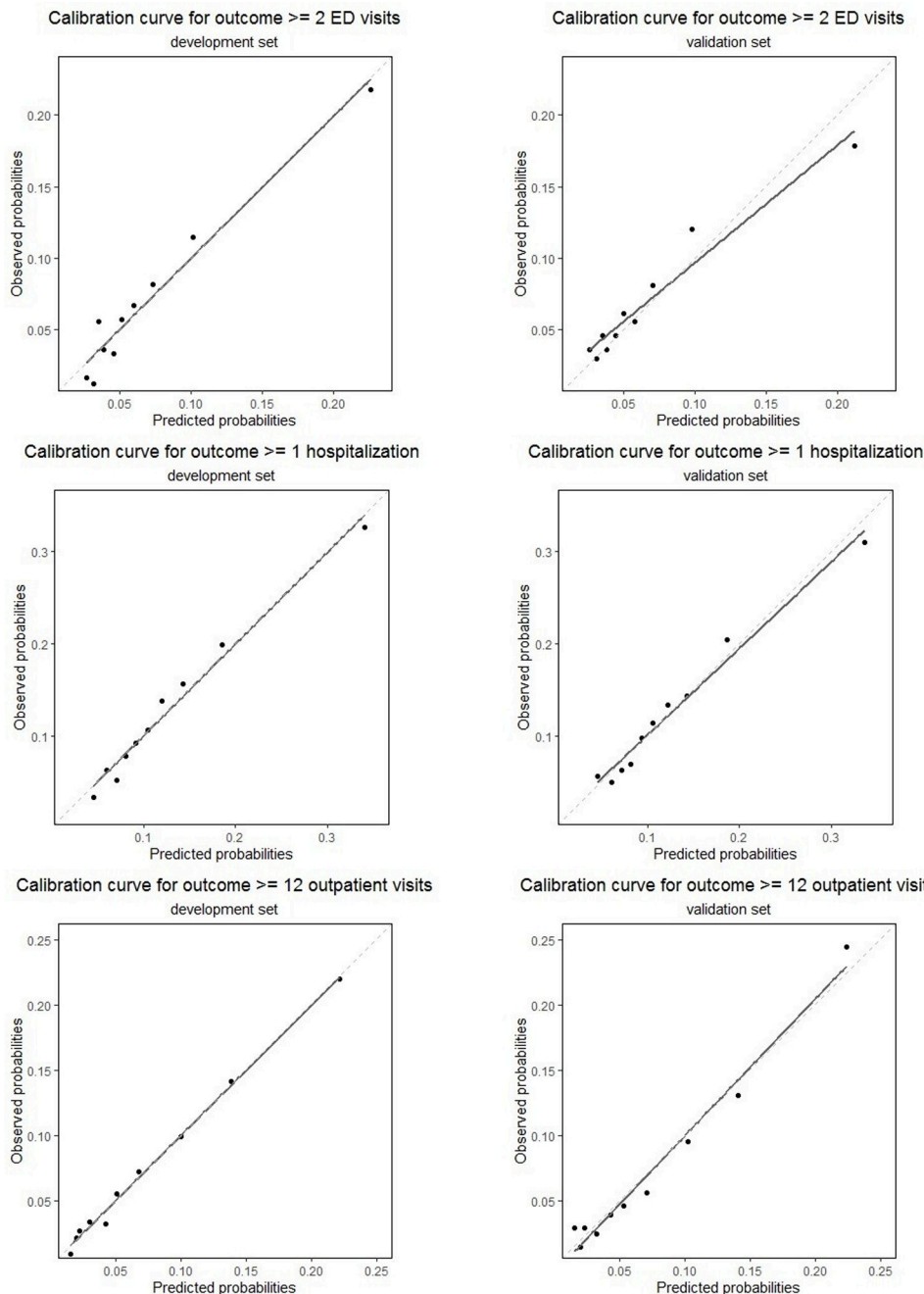

**Fig 1. Calibration curves for the three prediction models.** The models for hospitalization and ED visits overestimate the risk in patients with a higher predicted risk. The model for outpatient visits has a reasonable agreement between predicted and actual risk.

In all three of our models, age, ≥6 chronic/oncologic diagnoses, the number of ED visits, and a higher number of outpatient visits in the year prior were significant predictors of health-care utilization one year later. This is consistent with the acute care models with the best model accuracy described by Wallace et al. (2014), with age, prior healthcare utilization and a multimorbidity measure as some of the most important predictors [42]. However, we expected

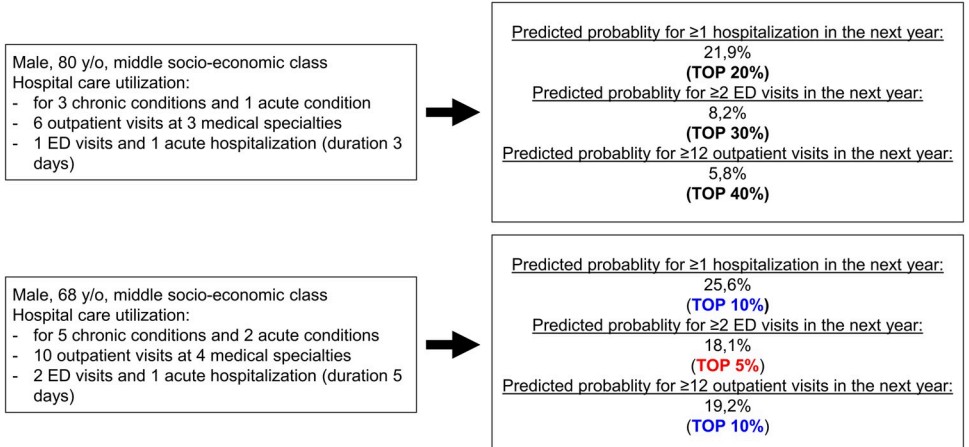

**Fig 2. Predicted probabilities for two fictive patients using the developed models and indicating the probability percentile per outcome.**

that the number of chronic diagnoses would have been a stronger predictor, based on the association between multimorbidity and healthcare utilization and healthcare costs reported in prior research [21,23]. Consistent with Heins et al. (2020), we found that higher numbers of outpatient visits in the prior year was the strongest predictor of higher numbers of outpatient visits one year later, and also predicted hospitalization and ED visits one year later [33]. These findings suggest that it is feasible to include measures of age, multimorbidity and prior healthcare utilization in models to predict future healthcare utilization.

A strength of our study is that we developed reasonably performing prediction models using local administrative data from the EHR. Prediction models based on national and regional data might perform worse in a local population and be less applicable due to local variations [42]. Our results suggest that administrative data from the local EHR are sufficient to develop reasonably performing prediction models. Another strength of our study is the inclusion of a large, general hospital population with multimorbidity to develop these prediction models. This population matches the general definition of multimorbidity [1]. The prediction models can aid healthcare professionals in the hospital in differentiating between several patients with multimorbidity in the general hospital population. The combination of variables such as age, number of chronic diagnoses and prior healthcare utilization and the associated risk for adverse outcomes can be used in addition to the clinical assessment. Another strength of our study is the interpretability of the models. Compared to black-box models, that tend to have the best performance, models such as multiple logistic regression generally have lower accuracy, but are more interpretable, which is beneficial for the usage in the clinical setting [45]. Models with high interpretability can offer insight into the relative importance of each predictor and can help to form hypotheses about how and why the model predicts high probability for certain patients. Further research using newer techniques with regularization and sample size calculation for the required events per candidate predictor might improve the accuracy of the models without losing interpretability [46,47]. Moreover, other interpretable models, like classification trees and random forest, may perform better if there are important interaction effects between predictor variables. A limitation of our prediction models is their overestimation of the actual probability for the higher risk groups, especially for the acute care models. However, if a higher predicted risk for healthcare utilization is considered an

indication of need for support, the overestimation could be acceptable if the models are used in combination with clinical assessment of the need for support.

Digitalization of health records and data generation gives rise to the possibility of integrating prediction models in the EHR and using recent EHR data and machine learning for automatic stratification of patients with multimorbidity at risk for adverse outcomes [26,48,49]. However, future impact studies should evaluate if patients who are identified as high risk patients are indeed patients with a high modifiable risk for these outcomes and if they would benefit from a more integrated care approach. Moreover, factors like perceived health status, coordination of care, health literacy or the reason for healthcare utilization are not included as variables in these models and not a standard part of EHR registration. In the future, further development and use of artificial intelligence solutions could aid in retrieving information that is not a standard part of EHR registration. Including these factors in the models or in the identification process might add valuable information about a patient's need for more integrated care [6,50]. Moreover, the models' performance and identification process could also be improved by adding more variables by connecting to and using data from other (local) data sources, e.g. mortality from the municipal database or number of general practitioner's visits from local general practitioners' databases [26]. Adding a prediction model for mortality to the identification process could be valuable, as the majority of patients approaching end of life are not being appropriately identified as such, and might also benefit from a more integrated care approach to enhance adequate advance care planning [51].

Our prediction models can be considered a useful example of how local prediction models could support individual healthcare professionals in the identification of high risk of hospitalization, emergency department visits, and outpatient visits in patients with multimorbidity. Hospitals could use their own administrative data, and our predictors for hospitalizations, emergency department visits and outpatient visits for patients with multimorbidity. By locally developing and validating these models, local variation of the hospital population will be taken into account. The development and internal validation of local prediction models could be the first step in developing an automated alert system in EHRs for identifying patients with multimorbidity who might benefit from an integrated care approach.

## Supporting information

**S1 Fig. Flow chart patients included in final dataset.**
(TIF)

**S2 Fig. Distribution of predicted probabilities in the development datasets based on the developed models, with cut-off values for top 5% and top 10% probabilities.**
(TIF)

**S1 Table. Comparison of development and validation data.** Total dataset (n = 18180) was split randomly three times, weighted for every outcome.
(PDF)

**S2 Table. Full Prognostic Model including intercept and model performance measures for derivation and validation set for outcome measure '≥1 hospitalization(s) in 2018'.**
(PDF)

**S3 Table. Full Prognostic Model including intercept and model performance measures for derivation and validation set for outcome measure '≥2 ED visits in 2018.**
(PDF)

**S4 Table. Full Prognostic Model including intercept and model performance measures for derivation and validation set for outcome measure '≥12 outpatient visits in 2018'.**
(PDF)

## Acknowledgments

We thank G. Klop and R. van de Kerkhof, data scientists, for their help with the data collection and the data cleaning process. We thank H. van der Zaag, MD, PhD and epidemiologist, for her collaboration.

## Author Contributions

**Conceptualization:** Marlies Verhoeff, Barbara C. van Munster.

**Data curation:** Marlies Verhoeff.

**Formal analysis:** Marlies Verhoeff.

**Funding acquisition:** Barbara C. van Munster.

**Investigation:** Marlies Verhoeff.

**Methodology:** Marlies Verhoeff, Janke de Groot, Barbara C. van Munster.

**Project administration:** Marlies Verhoeff.

**Resources:** Barbara C. van Munster.

**Supervision:** Janke de Groot, Jako S. Burgers, Barbara C. van Munster.

**Validation:** Marlies Verhoeff, Jako S. Burgers, Barbara C. van Munster.

**Visualization:** Marlies Verhoeff.

**Writing – original draft:** Marlies Verhoeff.

**Writing – review & editing:** Marlies Verhoeff, Janke de Groot, Jako S. Burgers, Barbara C. van Munster.

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
