## [Decision Letter · Decision Letter 0]

23 Aug 2021

PONE-D-21-09724

Predicting future hospital care utilization by patients with multimorbidity using electronic health record data

PLOS ONE

Dear Dr. Verhoeff,

Thank you for submitting your manuscript to PLOS ONE. After careful consideration, we feel that it has merit but does not fully meet PLOS ONE’s publication criteria as it currently stands. Therefore, we invite you to submit a revised version of the manuscript that addresses the points raised during the review process.

We look forward to receiving your revised manuscript.

Kind regards,

Ram Chandra Bajpai, Ph.D.

Academic Editor

PLOS ONE

Additional Editor Comments (if provided):

As it seems a model development study. Authors should include it in the title and clearly mention in the methods section.

Forward stepwise variable section is not recommended when building a prediction model as regression coefficients may be unstable and some important combination of predictors may be missed. Authors should use recommendations by Heinze et al 2018 for variable selection (https://onlinelibrary.wiley.com/doi/full/10.1002/bimj.201700067).

Patient selection process should be better represented by a flow diagram so authors should consider it.

Why did authors not formally calculated power for each outcome of interest? Authors should use Riley et al 2020 (https://www.bmj.com/content/368/bmj.m441) to demonstrate appropriateness of study power for each outcome.

Authors have used a weighted split-sample procedure for internal validation. This statement required a proper citation.

Authors should also consider some additional model calibration measures such as Brier score, and expected /observed event ratio etc.

Model algorithm (or final equation) for each outcome must be presented in the manuscript so others will know about how to calculate risk for a given patient.

Included figures are not clear and readable. Kindly add high resolution figures.

Journal Requirements:

2. Thank you for including your ethics statement:  "The local institutional review board104approved the anonymous use of thesedata for research purposesanda waiver of consent(Gelre LTC 105number2019_02)".   

3. Please correct your reference to "p=0.000" to "p<0.001" or as similarly appropriate, as p values cannot equal zero.

4. In your Methods section, please ensure that sufficient information to make the study reproducible are provided (for example, by describing the models and equations  used, and describing parameters and assumptions applied).

5. Please upload a new copy of Figure 1 as the detail is not clear. Please follow the link for more information: https://blogs.plos.org/plos/2019/06/looking-good-tips-for-creating-your-plos-figures-graphics/" https://blogs.plos.org/plos/2019/06/looking-good-tips-for-creating-your-plos-figures-graphics/.

Reviewers' comments:

Reviewer's Responses to Questions

**Comments to the Author**

1. Is the manuscript technically sound, and do the data support the conclusions?

Reviewer #1: Yes

Reviewer #2: Yes

2. Has the statistical analysis been performed appropriately and rigorously? 

Reviewer #1: Yes

Reviewer #2: Yes

3. Have the authors made all data underlying the findings in their manuscript fully available?

Reviewer #1: Yes

Reviewer #2: Yes

4. Is the manuscript presented in an intelligible fashion and written in standard English?

Reviewer #1: Yes

Reviewer #2: Yes

5. Review Comments to the Author

Reviewer #1: Thank you for the opportunity to review the submission of the manuscript entitled "Predicting future hospital care utilization by patients with multimorbidity using electronic health record data". The authors have written a very succinct and interesting manuscript, which will contribute to the knowledge base of multimorbidity. I have included suggested revisions below and would be happy to review a resubmission.

-Could the authors please further clarify why these separate groups of patients were created : "who had received outpatient clinical [care?] for two or more chronic diagnoses, two or more oncological diagnoses or at least one chronic and one oncological diagnosis in 2017 and who had received hospital care for at least one diagnosis in 2018"?

-Please change "multimorbid patients" to "patients with multimorbidity"

-Please change "probability for patient" as it is not clear what this sentence is describing

-Could the authors please clarify whether data about the reason(s) for hospitalizations, emergency department visits or outpatient visits were available?

-If this data were not available, what potential insight would this data provide and if this data were available, why was this data not included in the analyses?

Reviewer #2: This paper seeks to establish whether data from electronic health records can be used to predict future healthcare utilization as measured by >=1 acute hospital stays, >=2 ED visits, or >=12 outpatient visits. Numerous studies have been conducted on this topic, but this is unique in that it is focused on the population with multimorbidity, defined as 2 or more chronic conditions. The results show that with basic information from the EHR, future health utilzation can be predicted with mild accuracy (c-statistic around 0.70)

Overall, the manuscript was technically sound, and the results presented are appropriate. The sample size is large enough for the methods used. One strength is the authors used the train/testing approach to build their models on 2/3 of the data, and validate on the remaining one-third. The results presented in the manuscript and supplementary material provide a good deal of transparency, and the methods are described well enough to replicate the study.

Although the methods used are valid - they are not state-of-the-art when it comes to building models where the main goal is prediction (as opposed to inference). Linear Regularization methods like LASSO and Elastic-Net generally perform better at prediction than stepwise logistic models. Further, methods like classification trees and Random Forest, may do better if there are important interaction effects between predictor variables. The authors should address this in the limitations. Further, they may want to highlight some of the strengths of the model they used, namely that it is interpretable, compared to many other "black box" methods.

6. PLOS authors have the option to publish the peer review history of their article (what does this mean?). If published, this will include your full peer review and any attached files.

Reviewer #1: No

Reviewer #2: No

---

## [Author Response · Author response to Decision Letter 0]

4 Oct 2021

Editor 

Thank you very much for your time and considering our manuscript for publication. Moreover, thank you for your additional comments and suggestions, that we have addressed point by point below and in the Response to Reviewers-file. 

- As it seems a model development study. Authors should include it in the title and clearly mention in the methods section. 

Thank you for your suggestion, we have changed the title of our manuscript and added specification to our method. p1, line 4 p4, line 94

- Forward stepwise variable section is not recommended when building a prediction model as regression coefficients may be unstable and some important combination of predictors may be missed. Authors should use recommendations by Heinze et al 2018 for variable selection (https://onlinelibrary.wiley.com/doi/full/10.1002/bimj.201700067).

Thank you for your input and the provided literature. After careful consideration of the provided literature, it seems that there are two issues:

1. We realized that the term “stepwise” in some papers means the automated selection using an algorithm that determines in each step which variable is added to the model next. However, we did not use an automated algorithm for this selection, but evaluated the effect of each additional variable ourselves. Therefore, it might be less confusing when we use the term “forward selection” instead of “forward stepwise selection”. We have changed this term in our manuscript. P8, line 185

2. The choice for forward selection instead of backward selection. As we understand, Heinze et al (2018) prefer backward elimination over forward selection. The reference that Heinze et al (2018) used to support their preference is Mantel (1970). But in the same journal Baele (1970) already critically commented on Mantel’s ideas. There is still an ongoing debate about the best method for developing prediction models. To our knowledge, both forward selection (FS) and backward elimination (BE) have limitations. For example, FS can miss an important combination of predictors, whereas BE might discard an important predictor due to a “nonsense” correlation between variables (Baele (1970)). We decided on forward selection, mainly because our sample size was large enough and it provided us with the opportunity to see the additional effect of each new variable that was added to the model. This also gave us the opportunity to carefully evaluate whether collinearity was present, causing coefficients and standard errors to blow up. 

- Patient selection process should be better represented by a flow diagram so authors should consider it.

We understand that ‘plain text’ representation of the inclusion criteria might not be ideal. We have considered to present the data in a flow diagram, but feel that the information might be more readable as a list of bullets, which we added to the methods section. We hope you will agree. However, we also added a flow diagram as extra supplementary material (named S2_fig) to consider for publication (we have not yet added this figure to the manuscript). We leave the decision to the Editor. P 5, line 101-105

- Why did authors not formally calculated power for each outcome of interest? Authors should use Riley et al 2020 (https://www.bmj.com/content/368/bmj.m441) to demonstrate appropriateness of study power for each outcome. 

Thank you for your suggestion. We understand that Riley et al (2020) offer new insights into the events per variable or events per candidate predictor parameter (EPP) and supply a new method to calculate the EPP. When we designed this study, this article was not yet available, so we have used the commonly used rule of thumb ( at least 10/15) as mentioned by Heinze et al 2018) with a large margin (lowest number of EPV was 27). We have added a remark on sample size calculation to the discussion section with reference to Riley et al (2020). P20, line 332

- Authors have used a weighted split-sample procedure for internal validation. This statement required a proper citation. 

Thank you for your suggestions, we have added the required reference. P8, line 190

- Authors should also consider some additional model calibration measures such as Brier score, and expected /observed event ratio etc. 

Thank you for your suggestions. We chose the calibration curves as we feel that they give the best insight in the agreement between observed and expected probability. Our idea for the use of local prediction models for multimorbidity is to identify (groups of) patients with higher risks for the outcome compared to other patients. We did not expect to be able to perfectly predict whether or not patients will have the outcome as both multimorbidity and the causes for the outcomes are complex and influenced by many factors that could not be included using EHR data. Therefore, we feel that it is most important to show the agreement between observed and predicted probability. This enables users to assess how trustworthy the relatively higher risk is. 

- Model algorithm (or final equation) for each outcome must be presented in the manuscript so others will know about how to calculate risk for a given patient. 

Thank you for your suggestion. The aim of our research was to develop, validate and evaluate the performance of these prediction models based on EHR data. We show that it was possible to develop prediction models with mild accuracy with EHR data. It was not our primary aim to develop prediction models that will be used by others, but we do understand that others might want to use the models and/or check the external validity for example. For this purpose, we have provided the full models in the supplementary material, with the regression coefficients that can be used to calculate the risk for a given patient. The information for calculation available in the supplementary material could also be published in the main text. We leave this decision to the Editor. 

- Included figures are not clear and readable. Kindly add high resolution figures. 

Thank you for your comment, we have added figures with higher resolution, used PACE to check the requirements and hope they are clear and readable now. 

Journal requirements 

Thank you for your comment. We aimed to follow all PLOS ONE’s style requirements, but do apologize if we forgot any. Please let us know if any requirements are not met in the submission of the revised manuscript . 

- Thank you for including your ethics statement: "The local institutional review board104approved the anonymous use of thesedata for research purposesanda waiver of consent (Gelre LTC 105number2019_02)". Please amend your current ethics statement to include the full name of the ethics committee/institutional review board(s) that approved your specific study. Once you have amended this/these statement(s) in the Methods section of the manuscript, please add the same text to the “Ethics Statement” field of the submission form (via “Edit Submission”). 

Thank you for your comment, we have made the requested amendments. P5, line 107.

- Please correct your reference to "p=0.000" to "p<0.001" or as similarly appropriate, as p values cannot equal zero. 

Thank you for your comment, we have corrected our reference. All tables in manuscript.

- In your Methods section, please ensure that sufficient information to make the study reproducible are provided (for example, by describing the models and equations used, and describing parameters and assumptions applied) 

Thank you for your comment. As reviewer #2 states the methods are described well enough to replicate the study, we are uncertain about what information is missing. The full models with regression coefficients that can be used to calculate individual risk for a patient are added as supplementary tables. 

- Please upload a new copy of Figure 1 as the detail is not clear. 

Thank you for your comment, we have re-added figure 1 with higher resolution and hope it is clear and readable now. 

Reviewer #1 

- Thank you for the opportunity to review the submission of the manuscript entitled "Predicting future hospital care utilization by patients with multimorbidity using electronic health record data". The authors have written a very succinct and interesting manuscript, which will contribute to the knowledge base of multimorbidity. I have included suggested revisions below and would be happy to review a resubmission. 

Thank you very much for your time and positive feedback on our manuscript. Moreover, thank you for your comments and suggestions, that we address point by point below and in the Response to Reviewers-file. 

- Could the authors please further clarify why these separate groups of patients were created : "who had received outpatient clinical [care?] for two or more chronic diagnoses, two or more oncological diagnoses or at least one chronic and one oncological diagnosis in 2017 and who had received hospital care for at least one diagnosis in 2018"? 

Thank you for your question. When designing this study, we aimed to study a hospital population with multimorbidity. As described, multimorbidity is generally defined as two or more chronic conditions. In the Netherlands, the Clinical Classification Software diagnoses are categorized into acute, chronic, elective, oncological and other diagnoses. We had a discussion about the oncological diagnoses, because the clinicians in our research group felt that many oncological conditions/care over the years has turned into a specific type of chronic care. That is why we decided to include oncological diagnoses as chronic conditions. We have changed the description of this inclusion criterium in the methods to further clarify the inclusion. Furthermore, we only included patients who had received hospital care for at least one diagnosis in 2018, because we were interested in those patients that we might aim an intervention at to coordinate and tailor hospital care. P5, line 103-105

- Please change "multimorbid patients" to "patients with multimorbidity" 

Thank you for your suggestion. We have changed this text. P6, line 132

- Please change "probability for patient" as it is not clear what this sentence is describing

Thank you for your careful reading. It appears that there was an ‘s’ missing, and the sentence should have said “probability for patients with ..”. We have added the ‘s’ and hope the sentence is clear with this adjustment. P17, line 264

- Could the authors please clarify whether data about the reason(s) for hospitalizations, emergency department visits or outpatient visits were available?-If this data were not available, what potential insight would this data provide and if this data were available, why was this data not included in the analyses? 

Thank you for your questions. The data we used for the development of the models are data that are registered in the EHR for both medical record purposes and financial claim reasons. These are part of the so-called diagnosis and treatment combinations. This makes the data easily retrievable from the EHR. Reason(s) for hospitalizations, ED visits or outpatient visits are not required for financial claims to insurance. In general, data on reasons for healthcare use is registered as open text and thus more often contain missing data. However, we do agree that reasons for visiting the hospital could give valuable insight, such as the relationship to the presence of certain conditions, the severity of present diseases (e.g. many hospitalizations for specific condition or many outpatient visits for specific condition might suggest unstable disease). Further development and use of artificial intelligence solutions could aid in retrieving the reason(s) for healthcare utilization in the future. We have added a remark on the possible future use of AI to include information that is not a standard part of EHR registration to the discussion section. P20, line 347

Reviewer #2 

- This paper seeks to establish whether data from electronic health records can be used to predict future healthcare utilization as measured by >=1 acute hospital stays, >=2 ED visits, or >=12 outpatient visits. Numerous studies have been conducted on this topic, but this is unique in that it is focused on the population with multimorbidity, defined as 2 or more chronic conditions. The results show that with basic information from the EHR, future health utilzation can be predicted with mild accuracy (c-statistic around 0.70)

Overall, the manuscript was technically sound, and the results presented are appropriate. The sample size is large enough for the methods used. One strength is the authors used the train/testing approach to build their models on 2/3 of the data, and validate on the remaining one-third. The results presented in the manuscript and supplementary material provide a good deal of transparency, and the methods are described well enough to replicate the study. 

Thank you very much for your time and positive feedback. Moreover, thank you for your comment and suggestions, that we address point by point below and in the Response to Reviewers-file. 

- Although the methods used are valid - they are not state-of-the-art when it comes to building models where the main goal is prediction (as opposed to inference). Linear Regularization methods like LASSO and Elastic-Net generally perform better at prediction than stepwise logistic models. Further, methods like classification trees and Random Forest, may do better if there are important interaction effects between predictor variables. The authors should address this in the limitations. Further, they may want to highlight some of the strengths of the model they used, namely that it is interpretable, compared to many other "black box" methods. 

Thank you for your comment and the useful suggestions. We have added a paragraph to the discussion to address the points that you raised. P19-20, line 325-333

---

## [Decision Letter · Decision Letter 1]

18 Nov 2021

Development and internal validation of prediction models for future hospital care utilization by patients with multimorbidity using electronic health record data

PONE-D-21-09724R1

Dear Dr. Verhoeff,

We’re pleased to inform you that your manuscript has been judged scientifically suitable for publication and will be formally accepted for publication once it meets all outstanding technical requirements.

Kind regards,

Ram Chandra Bajpai, Ph.D.

Academic Editor

PLOS ONE

Additional Editor Comments (optional):

Reviewers' comments:

Reviewer's Responses to Questions

**Comments to the Author**

1. If the authors have adequately addressed your comments raised in a previous round of review and you feel that this manuscript is now acceptable for publication, you may indicate that here to bypass the “Comments to the Author” section, enter your conflict of interest statement in the “Confidential to Editor” section, and submit your "Accept" recommendation.

Reviewer #1: All comments have been addressed

Reviewer #2: All comments have been addressed

2. Is the manuscript technically sound, and do the data support the conclusions?

Reviewer #1: Yes

Reviewer #2: Yes

3. Has the statistical analysis been performed appropriately and rigorously? 

Reviewer #1: Yes

Reviewer #2: Yes

4. Have the authors made all data underlying the findings in their manuscript fully available?

Reviewer #1: Yes

Reviewer #2: No

5. Is the manuscript presented in an intelligible fashion and written in standard English?

Reviewer #1: Yes

Reviewer #2: Yes

6. Review Comments to the Author

Reviewer #1: Thanks very much for your responses and revisions based on reviewer feedback -- I believe that this manuscript is acceptable for publication.

Reviewer #2: (No Response)

7. PLOS authors have the option to publish the peer review history of their article (what does this mean?). If published, this will include your full peer review and any attached files.

Reviewer #1: No

Reviewer #2: No

---

## [Editor Report · Acceptance letter]

29 Nov 2021

PONE-D-21-09724R1 

Development and internal validation of prediction models for future hospital care utilization by patients with multimorbidity using electronic health record data 

Dear Dr. Verhoeff:

I'm pleased to inform you that your manuscript has been deemed suitable for publication in PLOS ONE. Congratulations! Your manuscript is now with our production department. 

Kind regards, 

on behalf of

Dr. Ram Chandra Bajpai 

Academic Editor

PLOS ONE